# 3D Off-Grid Localization for Adjacent Cavitation Noise Sources Using Bayesian Inference

**DOI:** 10.3390/s23052628

**Published:** 2023-02-27

**Authors:** Minseuk Park, Sufyan Ali Memon, Geunhwan Kim, Youngmin Choo

**Affiliations:** 1Department of Defense Systems Engineering, Sejong University, Seoul 05006, Republic of Korea; 2Department of Ocean Systems Engineering, Sejong University, Seoul 05006, Republic of Korea

**Keywords:** incipient cavitation, adjacent noise sources, sparse localization, off-grid

## Abstract

The propeller tip vortex cavitation (TVC) localization problem involves the separation of noise sources in proximity. This work describes a sparse localization method for off-grid cavitations to estimates their precise locations while keeping reasonable computational efficiency. It adopts two different grid (pairwise off-grid) sets with a moderate grid interval and provides redundant representations for adjacent noise sources. To estimate the position of the off-grid cavitations, a block-sparse Bayesian learning-based method is adopted for the pairwise off-grid scheme (pairwise off-grid BSBL), which iteratively updates the grid points using Bayesian inference. Subsequently, simulation and experimental results demonstrate that the proposed method achieves the separation of adjacent off-grid cavitations with reduced computational cost, while the other scheme suffers from a heavy computational burden; for the separation of adjacent off-grid cavitations, the pairwise off-grid BSBL took significantly less time (29 s) compared with the time taken by the conventional off-grid BSBL (2923 s).

## 1. Introduction

The propeller cavitation phenomenon is a critical factor that induces shipping noise [1]; thus, inferring the precise position of the cavitation noise source is required to reduce the shipping noise in the design stage of a vessel. In the incipient stage of cavitation, microbubbles are incepted into the propeller tip vortex line and developed to the tip vortex cavitation (TVC), which are usually invisible because they are too small to be observed [2,3]. To track the incipient TVC, acoustical approaches [2,3,4,5,6,7,8,9,10,11,12] are practically desirable rather than optical approaches [13,14].

Among the acoustical approaches, many related works have utilized virtual noise sources (referred to as potential noise sources) to estimate the cavitation locations [3,4,5,6,7,8,9,10,11]. For example, matched field processing (MFP)-based methods [3,4,5,6] exploit evenly distributed virtual noise sources to calculate the cavitation locations. In this process, the similarities between the pressure field induced by virtual noise sources (replica) and the acoustic measurement are calculated for each virtual noise source, and the virtual noise source with the maximum similarity is considered as the cavitation location. The MFP-based methods outperformed the time difference of the arrival-based methods [12] in terms of robustness to noise and, accordingly, have been widely used in cavitation localization problems [3,4,5,6].

However, MFP-based methods exhibited limited resolution when cavitation noise sources were adjacent or correlated. Recently, many related works adopted sparse reconstruction-based methods to overcome the low resolution of the MFP-based methods [7,8]. Sparse reconstruction, such as compressive sensing (CS) [15] and sparse Bayesian learning (SBL) [16], is a signal-processing method that efficiently solves the underdetermined system. In underwater acoustics, sparse reconstruction has been widely utilized to solve problems, such as beamforming [17,18,19,20], geoacoustic inversion [21], near-field acoustic holography [22,23], and source localization [24]. For cavitation localization, the sparse reconstruction achieved the high-resolution localization performance on the assumption that sparse cavitation noise sources lie on the virtual noise sources [8,9,10]. However, actual cavitation noise sources are lying on off-grid points; the sparse reconstruction-based methods resulted in the modeling error owing to basis mismatch, and exhibited degraded performance in practical applications.

To resolve the problem induced by basis mismatch, off-grid sparse localization [25,26] is adopted in the sparse cavitation localization problem [11]. Off-grid localization assumes that the signal model for the off-grid cavitation noise can be approximated by a combination of an on-grid potential noise source and their first-order Taylor expansion. By adopting the approximated signal model (off-grid signal model), the off-grid sparse localization exhibited reduced modeling error owing to basis mismatch error and provided high-resolution cavitation localization results. In practical applications, the off-grid sparse localization was utilized to 1D [25,26,27], 2D DOA [28,29] estimation, and 3D source localization [11] problems and outperformed the on-grid sparse localization method with respect to resolution.

Although off-grid sparse localization provides high-resolution localization results, their performance depends on the tradeoff between localization accuracy and the computational complexity. The main reason is that the off-grid sparse localization requires redundant representations, which adopt finer grid points, for estimating the precise source position; however, this leads to heavy computational complexity, especially for 3D problems [11,27].

In this paper, we adopted the off-grid signal model which employ two different grid point sets (pairwise off-grid signal model) to localize the off-grid cavitation. With a moderate grid interval, the pairwise off-grid signal model provides redundant representations, as well as computational efficiency for the localization of adjacent noise sources. Similar to the off-grid BSBL method [11], we provide an SBL-based sparse reconstruction scheme for solving the pairwise off-grid signal model (pairwise off-grid BSBL).

In Section 2, we describe the off-grid BSBL method, which was utilized to localize the off-grid cavitation. Section 3 provides the pairwise off-grid BSBL scheme, which outperforms the off-grid BSBL with respect to resolution and computational efficiency. The proposed method was applied to simulated and experimental data to examine the performance compared with the conventional localization method (Section 4). Finally, Section 5 provides the conclusion and contributions of the study.

## 2. Off-Grid Broadband Signal Model for Cavitation Localization

TVC noise transmits a spherical waveform, having a broadband frequency spectrum. In this section, we described a signal model based on the off-grid BSBL [11] to estimate the position of TVC noise.

### 2.1. Off-Grid Narrowband Signal Model

TVC noise, which is regarded as a monopole type source [2,3], emits a spherical wave with a broadband spectrum. Let ${\tilde{y}}_{m}^{\left(f_{l}\right)}$ be a received signal from a TVC noise source, which represents a complex signal at frequency $f_{l}$ measured by the *m*th hydrophone. By using the spherical wave propagation model [6], the measured sound pressure ${\tilde{y}}_{m}^{\left(f_{l}\right)}$ can be formulated as follows:(1)$${\tilde{y}}_{m}^{\left(f_{l}\right)}=\frac{e^{-j\left(2\pi f_{l}/c\right)r_{m}}}{4\pi r_{m}}{\tilde{x}}^{\left(f_{l}\right)}+{\tilde{n}}_{m}^{\left(f_{l}\right)},$$
where c is a sound speed in water, $r_{m}$ is the distance from the source to the *m*th hydrophone, ${\tilde{x}}^{\left(f_{l}\right)}$ is the complex source amplitude, and ${\tilde{n}}_{m}^{\left(f\right)}$ represents the noise involving electric noise, experimental error, and deviation of the acoustic model from actual sound propagation in the water medium [30].

To determine the source position, virtual sources and the search space are exploited here. Virtual sources are distributed at uniform grid points ${\left\{\left(u_{n},\;v_{n},\;w_{n}\right)\right\}}_{n=1}^{N}$ in the search space of a 𝐮-𝐯-**w** Cartesian coordinate system. When the actual cavitation noise sources lie on virtual sources without mismatch, the measured sound pressure can be derived via the superposition of sound pressures induced by virtual sources as follows [7]:(2)$${\tilde{y}}_{m}^{\left(f_{l}\right)}=\overset{N}{\underset{n=1}{\sum }}\frac{e^{-j\left(2\pi f_{l}/c\right)r_{m,n}}}{4\pi r_{m,n}}{\tilde{x}}_{n}^{\left(f_{l}\right)}+{\tilde{n}}_{m}^{\left(f_{l}\right)}=\overset{N}{\underset{n=1}{\sum }}{\tilde{a}}_{m,n}^{\left(f_{l}\right)}{\tilde{x}}_{n}^{\left(f_{l}\right)}+{\tilde{n}}_{m}^{\left(f_{l}\right)},$$
where ${\tilde{x}}_{n}^{\left(f_{l}\right)}$ represents the complex amplitude of the 𝑛th potential noise source, $r_{m,n}$ represents the distance from the 𝑛th potential noise source to the 𝑚th hydrophone, and 𝑁 represents the number of grid points (the number of virtual sources); ${\tilde{a}}_{m,n}^{\left(f_{l}\right)}$ reflects the sound propagation from the 𝑛th potential noise source to the 𝑚th hydrophone: ${\tilde{a}}_{m,n}^{\left(f_{l}\right)}=e^{-j\left(2\pi f_{l}/c\right)r_{m,n}}$.

Then, the relationship between the *M*-hydrophones (in the array) and *N*-potential noise sources (in the search space) can be expressed in vector form as follows [7]:(3)$${\tilde{y}}^{\left(f_{l}\right)}=\overset{N}{\underset{n=1}{\sum }}{\tilde{a}}_{n}^{\left(f_{l}\right)}{\tilde{x}}_{n}^{\left(f_{l}\right)}+{\tilde{n}}^{\left(f_{l}\right)},$$
where ${\tilde{y}}^{\left(f_{l}\right)}\in C^{M}$ is the measurement vector for the array, ${\tilde{n}}^{\left(f_{l}\right)}\in C^{M}$ is the noise vector, and ${\tilde{a}}_{n}^{\left(f_{l}\right)}\in C^{M}$ is the transformation vector for the 𝑛th potential noise source. In this system, only replicas corresponding to the true cavitation noise sources exhibit non-zero pressures. Therefore, the cavitation positions can be determined by solving the linear system of Equation (2). While finding the solution ${\tilde{x}}_{n}^{\left(f_{l}\right)}$, a normalization process is conducted which makes unit transformation vectors ($a_{n}^{\left(f\right)}$) to avoid the bias arising from different norms of the replicas [10]. Then, the normalized linear system of Equation (2) can be expressed as follows [7]:(4)$$\frac{{\tilde{y}}^{\left(f_{l}\right)}}{∥{\tilde{y}}^{\left(f_{l}\right)}{∥}_{2}}=\overset{N}{\underset{n=1}{\sum }}\frac{{\tilde{a}}_{n}^{\left(f_{l}\right)}}{∥{\tilde{a}}_{n}^{\left(f_{l}\right)}{∥}_{2}}x_{n}^{\left(f_{l}\right)}+\frac{{\tilde{n}}^{\left(f_{l}\right)}}{∥{\tilde{y}}^{\left(f_{l}\right)}{∥}_{2}}.$$

Denoting $y^{\left(f\right)}=\frac{{\tilde{y}}^{\left(f_{l}\right)}}{∥{\tilde{y}}^{\left(f_{l}\right)}{∥}_{2}},$ $a_{n}^{\left(f\right)}=\frac{{\tilde{a}}_{n}^{\left(f_{l}\right)}}{∥{\tilde{a}}_{n}^{\left(f_{l}\right)}{∥}_{2}}$, and $n^{\left(f\right)}=\frac{{\tilde{n}}^{\left(f_{l}\right)}}{∥{\tilde{y}}^{\left(f_{l}\right)}{∥}_{2}}$, Equation (4) can be simplified as follow [7]:(5)$$y^{\left(f_{l}\right)}=\overset{N}{\underset{n=1}{\sum }}a_{n}^{\left(f_{l}\right)}x_{n}^{\left(f_{l}\right)}+n^{\left(f_{l}\right)}.$$

Equation (5) represents narrowband signal model (also referred as on-grid narrowband signal model), which can efficiently estimate the source position via sparse reconstruction processes when cavitation sources lie on potential noise source positions. In practice, however, this signal model involves the modeling error, since actual cavitations do not lie on a discrete basis. To compensate the modeling error, an off-grid signal model [11] is introduced to localize the off-grid cavitation by adopting the off-grid point set ${\left\{\left(u_{n}+\Delta u_{n},\;v_{n}+\Delta v_{n},w_{n}+\Delta w_{n}\right)\right\}}_{n=1}^{N}$. Here, ${\left\{\left(\Delta u_{n},\;\Delta v_{n},\;\Delta w_{n}\right)\right\}}_{n=1}^{N}$ denotes the basis mismatches, which are deviations of the cavitation noise from the nearest grid point.

To derive the off-grid signal model, we assumed that the off-grid replicas can be approximated from the first-order Taylor expansion of ${\left\{a_{n}^{\left(f_{l}\right)}\right\}}_{n=1}^{N}$, which yields the following [11]:(6)$$y^{\left(f_{l}\right)}=\overset{N}{\underset{n=1}{\sum }}\left(a_{n}^{\left(f_{l}\right)}+{\frac{a_{n}^{\left(f_{l}\right)}}{\partial u}|}_{\left(u_{n},\;v_{n},\;w_{n}\right)}\cdot \Delta u_{n}+{\frac{a_{n}^{\left(f_{l}\right)}}{\partial v}|}_{\left(u_{n},\;v_{n},\;w_{n}\right)}\cdot \Delta v_{n}+{\frac{a_{n}^{\left(f_{l}\right)}}{\partial w}|}_{\left(u_{n},\;v_{n},\;w_{n}\right)}\cdot \Delta w_{n}\right)x_{n}^{\left(f_{l}\right)}+n^{\left(f_{l}\right)}.$$

Denoting $\Delta u={\left[\Delta u_{1},\cdots ,\Delta u_{n}\right]}^{T}$, $\Delta v={\left[\Delta v_{1},\cdots ,\Delta v_{n}\right]}^{T}$, and $\Delta w={\left[\Delta w_{1},\cdots ,\Delta w_{n}\right]}^{T}$, Equation (6) can be re-expressed as the following off-grid narrow-band signal model [11]:(7)$$\begin{array}{ll} y^{\left(f_{l}\right)} & =\left(A^{\left(f_{l}\right)}+D_{u}^{\left(f_{l}\right)}\mathrm{diag}\left(\Delta u\right)+D_{v}^{\left(f_{l}\right)}\mathrm{diag}\left(\Delta v\right)+D_{w}^{\left(f_{l}\right)}\mathrm{diag}\left(\Delta w\right)\right)x^{\left(f_{l}\right)}+n^{\left(f_{l}\right)}\\ & =\Omega^{\left(f_{l}\right)}x^{\left(f_{l}\right)}+n^{\left(f_{l}\right)}, \end{array}$$
where
(8)$$\begin{array}{l} A^{\left(f_{l}\right)}=\left[a_{1}^{\left(f_{l}\right)},\cdots ,a_{N}^{\left(f_{l}\right)}\right],\\ D_{u}^{\left(f\right)}=\left[{\frac{\partial a_{1}^{\left(f_{l}\right)}}{\partial u}|}_{\left(u_{1},\;v_{1},\;w_{1}\right)},\cdots ,{\frac{\partial a_{N}^{f_{l}}}{\partial u}|}_{\left(u_{N},\;v_{N},\;w_{N}\right)}\right],\\ D_{v}^{\left(f\right)}=\left[{\frac{\partial a_{1}^{\left(f_{l}\right)}}{\partial v}|}_{\left(u_{1},\;v_{1},\;w_{1}\right)},\cdots ,{\frac{\partial a_{N}^{\left(f_{l}\right)}}{\partial v}|}_{\left(u_{N},\;v_{N},\;w_{N}\right)}\right],\\ D_{w}^{\left(f\right)}=\left[{\frac{\partial a_{1}^{\left(f_{l}\right)}}{\partial w}|}_{\left(u_{1},\;v_{1},\;w_{1}\right)},\cdots ,{\frac{\partial a_{N}^{\left(f_{l}\right)}}{\partial w}|}_{\left(u_{N},\;v_{N},\;w_{N}\right)}\right]. \end{array}$$

Herein, $\Omega^{\left(f_{l}\right)}=\left[\Omega_{1}^{\left(f_{l}\right)},\cdots ,\Omega_{N}^{\left(f_{l}\right)}\right]$ is an off-grid transformation matrix and $\Omega_{n}^{\left(f_{l}\right)}$ is an off-grid transformation vector for the *n*th potential noise source (i.e., $\Omega_{n}^{\left(f_{l}\right)}=a_{n}^{\left(f\right)}+{\frac{a_{n}^{\left(f\right)}}{\partial u}|}_{\left(u_{n},\;v_{n},\;w_{n}\right)}\cdot \Delta u_{n}+{\frac{a_{n}^{\left(f\right)}}{\partial v}|}_{\left(u_{n},\;v_{n},\;w_{n}\right)}\cdot \Delta v_{n}+{\frac{a_{n}^{\left(f\right)}}{\partial w}|}_{\left(u_{n},\;v_{n},\;w_{n}\right)}\cdot \Delta w_{n}).$

### 2.2. Off-Grid Cavitation Signal Model

By adopting the broadband approach [11], the off-grid narrowband signal model can be extended to the broadband case. Before describing the off-grid broadband signal model, the off-grid narrowband systems over the frequencies ($f_{1},\cdots ,f_{L})$ can be denoted as the following systems:(9)$${\left\{y^{\left(f_{l}\right)}=\Omega^{\left(f_{l}\right)}x^{\left(f_{l}\right)}+n^{\left(f_{l}\right)}\right\}}_{l=1}^{L}.$$

In these systems, the potential noise sources, corresponding to the true cavitation noise sources, exhibit non-zero complex amplitudes over all frequencies (e.g., xn(fl)≠0 for ∀l); thus [x(f1),⋯,x(fL)] should be row-wise block-sparse (referred to as spectrally joint sparse [10,11]). To handle the characteristic of the block-sparse, we employ the coherent frequency signal model [10,11] which can be derived via the following process:(10)∑l=1Ly(fl)⊗el=∑l=1L(Ω(fl)x(fl)+n(fl))⊗el =[∑l=1LΩ(fl)⊗diag(el)]⋅[∑l=1Lx(fl)⊗el]+∑l=1Ln(fl)⊗el, 
where the operator ⊗ represents the Kronecker product, and el ∈ℝL represents basis vector (e.g., e1=[1,0,⋯,0]T).

By denoting $\Omega =\overset{L}{\underset{l=1}{\sum }}\left(\Omega^{\left(f_{l}\right)}⊗\mathrm{diag}\left(e_{l}\right)\right)$, $x=\overset{L}{\underset{l=1}{\sum }}\left(x^{\left(f_{l}\right)}⊗e_{l}\right)$, and $n=\overset{L}{\underset{l=1}{\sum }}\left(n^{\left(f_{l}\right)}⊗e_{l}\right)$, Equation (9) can be simply extended to a linear broadband system as follow [11]:(11)$$\begin{array}{l} y=\left(A+D_{u}\cdot \mathrm{diag}\left(\Delta u⊗e_{L}\right)+D_{v}\cdot \mathrm{diag}\left(\Delta v⊗e_{L}\right)+D_{w}\cdot \mathrm{diag}\left(\Delta w⊗e_{L}\right)\right)x+n\\ =\Omega x+n, \end{array}$$

where eL∈ℝL represents the all-ones vector, having 𝐿 components, and the narrowband deviations (Δu,Δv,Δw) are extended to (Δu⊗eL,Δv⊗eL,Δw⊗eL), since the deviations are the same over the frequencies.

Accordingly, the solution x is expressed as follows [10]:(12)$$x=\left[x_{1}^{\mathrm{block}};\cdots ;x_{N}^{\mathrm{block}}\right].$$

An element $x_{n}^{\mathrm{block}}={\left[x_{n}^{\left(f_{1}\right)},\cdots ,x_{n}^{\left(f_{L}\right)}\right]}^{T}$ comprises broadband amplitudes of the 𝑛th potential noise source; thus, x has a block-wise structure. Subsequently, remaining notations $y,A,\;\mathrm{and}\;D_{u,v,w}$ can be expressed as the following block-wise structures:(13)$$\begin{array}{l} y=\left[y_{1}^{\mathrm{block}};\cdots ;y_{M}^{\mathrm{block}}\right],\;\;\;A=\left[\begin{array}{ccc} A_{1,1}^{\mathrm{block}} & \cdots & A_{1,N}^{\mathrm{block}}\\ \vdots & \ddots & \vdots \\ A_{M,1}^{\mathrm{block}} & \cdots & A_{M,N}^{\mathrm{block}} \end{array}\right]\\ D_{u}=\left[D_{u_{:,1}}^{\mathrm{block}},\cdots ,D_{u_{:,N}}^{\mathrm{block}}\right]=\left[{\frac{\partial A_{:,1}^{\mathrm{block}}}{\partial u}|}_{\left(u_{1},\;v_{1},\;w_{1}\right)},\cdots ,{\frac{\partial A_{:,N}^{\mathrm{block}}}{\partial u}|}_{\left(u_{N},\;v_{N},\;w_{N}\right)}\right],\\ D_{v}=\left[D_{v_{:,1}}^{\mathrm{block}},\cdots ,D_{v_{:,N}}^{\mathrm{block}}\right]=\left[{\frac{\partial A_{:,1}^{\mathrm{block}}}{\partial v}|}_{\left(u_{1},\;v_{1},\;w_{1}\right)},\cdots ,{\frac{\partial A_{:,N}^{\mathrm{block}}}{\partial v}|}_{\left(u_{N},\;v_{N},\;w_{N}\right)}\right],\\ D_{w}=\left[D_{w_{:,1}}^{\mathrm{block}},\cdots ,D_{w_{:,N}}^{\mathrm{block}}\right]=\left[{\frac{\partial A_{:,1}^{\mathrm{block}}}{\partial w}|}_{\left(u_{1},\;v_{1},\;w_{1}\right)},\cdots ,{\frac{\partial A_{:,N}^{\mathrm{block}}}{\partial w}|}_{\;\left(u_{N},\;v_{N},\;w_{N}\right)}\right]. \end{array}$$
where $y_{m}^{\mathrm{block}}={\left[y_{m}^{\left(f_{1}\right)},\cdots ,y_{m}^{\left(f_{L}\right)}\right]}^{T}$ comprises the *L*-frequency components (multiple measurements) at the 𝑚th hydrophone of the array, and y denotes a multiple-measurement vector for the array. $A_{m,n}^{\mathrm{block}}$ reflects the sound propagation from the 𝑛th potential noise source to the 𝑚th hydrophone over the frequencies, where $A_{m,n}^{\mathrm{block}}=\mathrm{diag}\left(a_{m,n}^{\left(f_{1}\right)},\cdots ,a_{m,n}^{\left(f_{L}\right)}\right)$. By using the notations ($\Delta U,\Delta V,\Delta W)=\left(\mathrm{diag}\left(\Delta u⊗e_{L}\right),\;\mathrm{diag}\left(\Delta v⊗e_{L}\right),\;\mathrm{diag}\left(\Delta w⊗e_{L}\right)\right)$**,** Equation (10) can be simply derived as the following off-grid cavitation signal model [11]:(14)$$y=\left(A+D_{u}\cdot \Delta U+D_{v}\cdot \Delta V+D_{w}\cdot \Delta W\right)x+n.\;$$

### 2.3. Stochastic Model for Off-Grid Cavitation Localization

To solve the unknown x and (Δu,Δv,Δw) in Equation (14), the off-grid block-sparse Bayesian learning (BSBL) method is exploited here (also referred to as off-grid BSBL) [11]. BSBL is a sparse reconstruction method that efficiently solves the underdetermined system when the solution x is block-wise sparse [31]. At the incipient stage of cavitation, the TVC is aperiodically measured with a broadband impulse signal; thus, solution x should be block-sparse (which means TVCs are spatially sparse), as described in Section 2.2.

To solve the block-sparse solution x, a stochastic model using Bayesian inference [32,33] is utilized. The prior and likelihood models of the block-sparse linear system in Equation (13) can be described as the following Gaussian distributions, respectively [11]:(15)$$p\left(x_{n}^{\mathrm{block}};\gamma_{n},B_{n}\right)∼\;CN\left(0,\gamma_{n}B_{n}\right),n=1,\cdots ,N.$$
(16)$$\begin{array}{ll} p\left(y|x;\sigma^{2},\Delta u,\Delta v,\Delta w\right) & =CN\left(\left(A+D_{u}\cdot \Delta U+D_{v}\cdot \Delta V+D_{w}\cdot \Delta W\right)x\;,\gamma_{n}B_{n}\right)\\ & ={\left({\pi \sigma }^{2}\right)}^{-ML}\exp\left\{-\frac{1}{\sigma^{2}}∥y-\left(A+D_{u}\cdot \Delta U+D_{v}\cdot \Delta V+D_{w}\cdot \Delta W\right)x{∥}_{2}^{2}\right\} \end{array}$$
where $\gamma_{n}$ is an element of a hyperparameter that impose a block sparsity on x**,** 𝐁 is a correlation matrix, and $\sigma^{2}$ is a noise variance in the system. Then, the corresponding MAP estimation [11] for the off-grid broadband signal model can be expressed as follows [11]:(17)$$\hat{x}=\arg\underset{x}{\max}p\left(x|y;\phi \right),\;\;\;\mathrm{where}\;\phi ≜\left\{\gamma ,B,\sigma^{2},\Delta u,\Delta v,\Delta w\right\}.$$

Herein, ϕ denotes the nuisance parameters to be estimated, which determines the posterior of the off-grid cavitation system. By maximizing the posterior from Equation (17), we can obtain the optimal nuisance parameters, as follows [11]:(18)$$\hat{\phi }=\arg\underset{\phi }{\max}E_{x|y}\left\{p\left(x|y;\phi \right)\right\}.$$

This stochastic model provides an efficient way to estimate the optimal nuisance parameters $\hat{\phi }$, which is required for obtaining the solution $\hat{x}$. After the estimation process, therefore, we can obtain the unknown $\hat{x}$ and (Δu,Δv,Δw), which indicate the amplitude and position of the off-grid cavitation.

## 3. Pairwise Off-Grid BSBL for Cavitation Localization

The 3D off-grid BSBL [11] outperformed on-grid sparse localization methods [7,8,9,10] with respect to resolution; however, it exhibited inaccurate localization results when adjacent sources were present. Although the finer grid is desirable to improve the resolution in this case, it significantly increases the computational burden. This section introduces the pairwise off-grid BSBL framework to overcome the limitations of the 3D off-grid BSBL.

### 3.1. Two Different Grid Sets: Pairwise Off-Grid

Instead of using the fine grid points, we employed two different grid sets to represent the adjacent noise sources (referred to as pairwise grid sets in this work). We will show that these pairwise grid sets, which exploit the moderate grid interval, can provide high-resolution localization results for the adjacent noise sources while mitigating the computational burden efficiently.

Consider a search space that has the region governed by $u\in \left[u_{0},\;u_{0}+N_{1}d\right],\;v\in \left[v_{0},\;v_{0}+N_{2}d\right],\mathrm{and}\;w\in \left[w_{0},\;w_{0}+N_{3}d\right]$. To allocate the virtual noise sources, we exploit two different grid point sets, having a uniform distribution at grid interval *d*. Let the first grid set $P=\left[p_{1},\cdots ,p_{N}\right]$ have $N\left(=N_{1}N_{2}N_{3}\right)$-grid points that are evenly distributed in the search space. For convenience, we employ the notation (*i, j, k*) to allocate the grid points, which indicate the directional index for each *u*, *v*, and *w*-axis, as shown in Figure 1a. The grid number n∈[1,N] for the first grid set is defined as $n=i+\left(j-1\right)N_{1}+\left(k-1\right)N_{1}N_{2}$, and then their positions are expressed as follows:(19)$$p_{n}^{\left(0\right)}={\left[u_{n}^{\left(0\right)},v_{n}^{\left(0\right)},w_{n}^{\left(0\right)}\right]}^{T}={\left[u_{0}+d\left(i-0.5\right),\;v_{0}+d\left(j-0.5\right),w_{0}+d\left(k-0.5\right)\right]}^{T},$$
where superscript (0) denotes the initial stage of the iterative process and ($u_{0},v_{0},w_{0})$ is the reference location. The second grid set $P^{\prime }=\left[p_{N+1},\cdots ,p_{N+N^{\prime }}\right]$ has the evenly distributed $N^{\prime }\left(=\left(N_{1}+1\right)\left(N_{2}+1\right)\left(N_{3}+1\right)\right)$-grid points (Figure 1b). We employed another directional index (*i’, j’, k’*) to distinguish it from the first grid set. Herein, the grid number $n\in \left[N+1,N+N^{\prime }\right]$ is defined as $n=N+i^{\prime }+\left(j^{\prime }-1\right)\left(N_{1}+1\right)+\left(k^{\prime }-1\right)\left(N_{1}+1\right)\left(N_{2}+1\right)$, and their locations are expressed as follows:(20)$${p^{\prime }}_{n}^{\left(0\right)}={\left[u_{n}^{\left(0\right)},v_{n}^{\left(0\right)},w_{n}^{\left(0\right)}\right]}^{T}={\left[u_{0}+d\left(i^{\prime }-1\right),\;v_{0}+d\left(j^{\prime }-1\right),w_{0}+d\left(k^{\prime }-1\right)\right]}^{T}.$$

Denoting the initial pairwise grid sets as $\left[P^{\left(0\right)},P^{\left(0\right)}\prime \right]$, we can obtain the initial virtual noise sources lying on these grid points (Figure 1c). Consequently, the pairwise grid set [$P^{\left(0\right)},P^{\left(0\right)}\prime ]$ fills up the search space with ($N+N^{\prime })$-virtual noise sources, and the corresponding transformation matrix A (Equation (13)) can be expressed as follows:(21)$$A=A\left(\left[P^{\left(0\right)},P\prime^{\left(0\right)}\right]\right)=\left[\left[\begin{array}{ccc} A_{1,1}^{block} & \cdots & A_{1,N}^{block}\\ \vdots & \ddots & \vdots \\ A_{M,1}^{block} & \cdots & A_{M,N}^{block} \end{array}\right],\left[\begin{array}{ccc} A_{1,N+1}^{block} & \cdots & A_{1,N+N^{\prime }}^{block}\\ \vdots & \ddots & \vdots \\ A_{M,N+1}^{block} & \cdots & A_{M,N+N^{\prime }}^{block} \end{array}\right]\right]$$

### 3.2. Grid Update and Bounds

From Equation (21), we obtained the initial virtual noise sources to express the off-grid cavitations. In the next, the locations of pairwise grid points should be iteratively refined to represent the off-grid cavitation according to an update rule. This section illustrates how pairwise grid points are updated to converge to off-grid cavitations and estimate their positions.

Once deviations ($\Delta u_{n},\Delta v_{n},\Delta w_{n})$ are estimated from the stochastic model in Equation (18), the pairwise 3D off-grid BSBL updates the grid points (i.e., updates the bases) for each iteration according to the following equation:(22)$${\hat{u}}_{n}=u_{n}+\Delta u_{n},\;\;{\hat{v}}_{n}=v_{n}+\Delta v_{n},\;\;{\hat{w}}_{n}=w_{n}+\Delta w_{n}$$
where (${\hat{u}}_{n},{\hat{v}}_{n},{\hat{w}}_{n}$) represents the updated location, and $\left(u_{n},v_{n},w_{n}\right)$ and $\left(\Delta u_{n},\Delta v_{n},\Delta w_{n}\right)$ represent the location and the deviation of the *n*th grid point in the previous iteration step, respectively. In this update process, each grid point has upper and lower bounds, which prevent the divergence of the bases, as follows:(23)$${\hat{u}}_{n}\in \left[\;u_{n}{}^{-},u_{n}{}^{+}\right],{\hat{v}}_{n}\in \left[\;v_{n}{}^{-},v_{n}{}^{+}\right],\;\;{\hat{w}}_{n}\in \left[\;w_{n}{}^{-},w_{n}{}^{+}\right]$$
where ${\left(\cdot \right)}_{n}{}^{-}$ and ${\left(\cdot \right)}_{n}{}^{+}$ represent the lower and upper bounds for the *n*th updated grid, which are equivalent to the following:(24)$$\begin{array}{l} \left[u_{n}{}^{-}\;,u_{n}{}^{+}\right]=u_{n}^{\left(0\right)}+\left[-0.5d,\;0.5d\right]\\ \left[v_{n}{}^{-}\;,v_{n}{}^{+}\right]=v_{n}^{\left(0\right)}+\left[-0.5d,\;0.5d\right]\\ \left[w_{n}{}^{-}\;,w_{n}{}^{+}\right]=w_{n}^{\left(0\right)}+\left[-0.5d,\;0.5d\right] \end{array}$$

From Equations (23) and (24), each grid point belongs to the volume space (=$d^{3})$ as shown in (Figure 2a,b), refining its location for each iteration. Furthermore, two different volume sets induce the joint space set where two volume spaces are duplicated (Figure 2c), representing two adjacent off-grid cavitations jointly; therefore, the pairwise grid sets provide redundant representations regardless of the grid interval.

By employing the grid update in Equation (22) and bounds in Equation (23), we can obtain the following grid update rule: (25)$$\begin{array}{c} {\hat{u}}_{n}=\left\{\begin{array}{cc} u_{n}+\Delta u_{n} & \;for\;u_{n}+\Delta u_{n}\in \left[u_{n}{}^{-},u_{n}^{+}\right]\\ u_{n}^{-} & \;for\;u_{n}+\Delta u_{n}>u_{n}^{+}\\ u_{n}^{+} & \;for\;u_{n}+\Delta u_{n}<u_{n}{}^{-} \end{array}\right.,\\ {\hat{v}}_{n}=\left\{\begin{array}{cc} v_{n}+\Delta v_{n} & \;for\;v_{n}+\Delta v_{n}\in \left[v_{n}{}^{-},v_{n}^{+}\right]\\ v_{n}^{+} & \;for\;v_{n}+\Delta v_{n}>v_{n}{}^{+}\\ v_{n}^{-} & \;for\;v_{n}+\Delta v_{n}<v_{n}{}^{-} \end{array}\right.,\\ {\hat{w}}_{n}=\left\{\begin{array}{cc} w_{n}+\Delta w_{n} & for\;w_{n}+\Delta w_{n}\in \left[w_{n}{}^{-},w_{n}{}^{+}\right]\\ w_{n}{}^{-} & for\;w_{n}+\Delta w_{n}>w_{n}{}^{+}\\ w_{n}{}^{-} & for\;w_{n}+\Delta w_{n}<w_{n}{}^{-} \end{array}\right.. \end{array}$$

For each iteration, the pairwise grid point sets are updated to be $\left[P\;P^{\prime }\right]\leftarrow {\left[\hat{u},\hat{v},\hat{w}\right]}^{T}$.

### 3.3. Grid Acvitation and Transfer

We found empirically that the pairwise off-grid BSBL sometimes fails to converge to the optimum solution due to the noisy condition or the limited number of hydrophones. To solve the problem, we contrive the grid activation and transfer rule.

By using the grid update process in Equation (25), the grid point is iteratively updated to approach the actual off-grid cavitation; however, it is sometimes captured in their bounds and fails to escape from the bounds (Figure 3a,b). In this case, grid points with a small $\left|\gamma_{n}\right|$ have little effect on the MAP process (Equation (18)), so there is no need to update all grid points. Here, we set a threshold to determine the active grid points (grid activation rule): the grid point $p_{n}$, which has a $\left|\gamma_{n}\right|$ larger than 0.05⋅max(|γ|), is activated and applied to the update rule in Equation (25), where $\gamma_{n}$ is a hyper-parameter of the *n*th entry, and max|γ| represent the maximum absolute value over all hyper-parameters $\gamma =\left[\gamma_{1},\cdots ,\gamma_{N}\right]$.

Subsequently, we define the grid transfer rule to prevent the local convergence under noisy conditions or with a limited number of hydrophones, which transfer the information of the virtual noise source to its neighbor grid points. Consider the two grid points $p_{n_{1}}$ and $p_{n_{2}}$ which belong to the same grid set and share a bound in a certain direction (Figure 3a). Assuming that only $p_{n_{1}}$ is activated and actual off-grid cavitation lies on the neighbor space, $p_{n_{1}}$ tends to converge to its bound after several iterations, which results in local convergence (Figure 3b). In the practical implementation, we set a loop to escape the convergence (grid transfer rule), as follows:(26)$$p_{n_{2}}=p_{n_{1}},\;\;p_{n_{1}}=p_{n_{1}}^{\left(0\right)},\;\gamma_{n_{2}}=\gamma_{n_{1}},\;\;\gamma_{n_{1}}=\frac{1}{L}\mathrm{Tr}\left[B^{-1}\left(\Sigma_{x,n_{2}}+\left(\mu_{n_{2}}^{\mathrm{block}}\right){\left(\mu_{n_{2}}^{\mathrm{block}}\right)}^{H}\right)\right],\;$$
where $p_{n_{1}}^{\left(0\right)}$ denotes the initial position of $n_{1}\mathrm{th}$ grid point. In Equation (26), $p_{n_{2}}$ inherits the information of the $p_{n_{1}}$ and becomes activated; meanwhile, $p_{n_{1}}$ becomes initialized and inactivated. After a few iterations, $p_{n_{2}}$, instead of $p_{n_{1}}$, approaches the actual off-grid cavitation and converges (Figure 3c), which result in accurate localization.

### 3.4. Estimating the Nuisance Parameters

We can obtain the optimal nuisance parameters $\hat{\phi }=\left\{\hat{\gamma },\hat{B},\;{\hat{\sigma }}^{2},\Delta \hat{u},\Delta \hat{v},\Delta \hat{w}\right\}$ required for the MAP in Equation (17) by using the EM (expectation–maximize) estimation, equivalent to maximizing the evidence 𝑝(𝐲|𝜙) and satisfying the following equation [11]:(27)$$\hat{\phi }=\arg\underset{\phi }{\max}E_{x|y;\phi }\left[\log p\left(y|x;\sigma^{2},\Delta u,\Delta v,\Delta w\right)p\left(x;\gamma ,B\right)\right]$$

By differentiating for each nuisance parameter [11], we can obtain the nuisance parameters as the following equations [11]:(28)$$\begin{array}{l} \gamma_{n}=\frac{1}{L}\mathrm{Tr}\left[B^{-1}\left(\Sigma_{x,n}+\left(\mu_{n}^{\mathrm{block}}\right){\left(\mu_{n}^{\mathrm{block}}\right)}^{H}\right)\right],\;\\ B=\frac{1}{N+N^{\prime }}\overset{N+N^{\prime }}{\underset{n=1}{\sum }}\frac{\Sigma_{x,n}+\left(\mu_{n}^{\mathrm{block}}\right){\left(\mu_{n}^{\mathrm{block}}\right)}^{H}}{\gamma_{n}}\;,\\ \sigma^{2}=\frac{∥y-\Phi \mu {∥}_{2}^{2}+\mathrm{Tr}\left(\Sigma_{x}A^{H}A\right)}{ML},\\ \Delta u={\left(R_{u,1}+R_{u,2}\right)}^{-1}\left(r_{u,1}+r_{u,2}\right),\\ \Delta v={\left(R_{v,1}+R_{v,2}\right)}^{-1}\left(r_{v,1}+r_{v,2}\right),\\ \Delta w={\left(R_{w,1}+R_{w,2}\right)}^{-1}\left(r_{w,1}+r_{w,2}\right), \end{array}$$
where $\mu_{n}^{\mathrm{block}}$ and $\Sigma_{x,n}$ denote the *n*th block in μ and $\Sigma_{x}$, respectively (herein, $\Sigma_{x}={\left(\sigma^{-2}A^{H}A+\Sigma_{0}^{-1}\right)}^{-1}\;,\;\mu =\left(\sigma^{-2}\Sigma_{x}A^{H}y\right)$, and $\Sigma_{0}=\mathrm{diag}\left\{\gamma_{1}B,\;\cdots ,\gamma_{N+N^{\prime }}B\;\right\}$). The corresponding derivation and notations (R and r) are detailed in Appendix A.

### 3.5. Localization Scheme Using Pairwise Off-Grid BSBL

In this section, we summarized the 3D pairwise off-grid cavitation localization procedure as the flow-chart (Figure 4) and the Algorithm 1.

**Algoritm 1.** 3D pairwise off-grid cavitation localization algorithm.1: Input: **y**, **A** and $D_{u,v,w}$ at pairwise grids [$P^{\left(0\right)}$ $P\prime^{\left(0\right)}$]    (Initial transformation matrix and derivative matrices)2: Initialization: $\gamma_{n}=1,\;\left(\forall n\in \left[1,N+N^{\prime }\right]\right),\;\sigma^{2}=0.001,\;\;k=1$
3: **while**
*k* < 2000 **do**4:    $\Sigma_{0}=\mathrm{diag}\left\{\gamma_{1}B,\;\cdots ,\gamma_{N+N^{\prime }}B\;\right\}$
5:    $\Sigma_{x}={\left(\sigma^{-2}A^{H}A+\Sigma_{0}^{-1}\right)}^{-1},\;\mu =\left(\sigma^{-2}\Sigma_{x}A^{H}y\right)$
6:    **for**
*n* = 1; *n* ≤ *N* + *N*^′^; *n*++ do7:         $\gamma_{n}=\frac{1}{L}\mathrm{Tr}\left[B^{-1}\left(\Sigma_{x,n}+\left(\mu_{n}^{\mathrm{block}}\right){\left(\mu_{n}^{\mathrm{block}}\right)}^{H}\right)\right]$
8:         $B=\frac{1}{N+N^{\prime }}\overset{N+N^{\prime }}{\underset{n=1}{\sum }}\frac{\Sigma_{x,n}+\left(\mu_{n}^{\mathrm{block}}\right){\left(\mu_{n}^{\mathrm{block}}\right)}^{H}}{\gamma_{n}}$
9:      **end for**10:    $\sigma^{2}=\frac{∥y-A\mu {∥}_{2}^{2}+\mathrm{Tr}\left(\Sigma_{x}A^{H}A\right)}{ML}$
11:    $\Delta u={\left(R_{u,1}+R_{u,2}\right)}^{-1}\left(r_{u,1}+r_{u,2}\right)$
12:    $\Delta v={\left(R_{v,1}+R_{v,2}\right)}^{-1}\left(r_{v,1}+r_{v,2}\right)$
13:    $\Delta w={\left(R_{w,1}+R_{w,2}\right)}^{-1}\left(r_{w,1}+r_{w,2}\right)$
14:    $\left[P\;P^{\prime }\right]\leftarrow {\left[\hat{u},\hat{v},\hat{w}\right]}^{T}$ (according to update and activation rule) 15:    re-estimate the $\left[P\;P^{\prime }\right],\;\gamma_{n}\;\;$(according to transfer rule) 16:    $A=A\left(\left[P\;P^{\prime }\right]\right),\;\;D_{u,v,w}=D_{u,v,w}\left(\left[P\;P^{\prime }\right]\right)$
17:    $\varepsilon =\gamma^{\mathrm{new}}-\gamma^{\mathrm{old}}{}_{2}∕\gamma^{\mathrm{old}}{}_{2}$, *k* = *k* + 118: **end while**19: $\hat{x}=\mu$
20: Output: $\hat{x}$, $\left[P\;P^{\prime }\right]$


## 4. Results

In this section, 3D pairwise off-grid BSBL was applied to simulated and measured data to validate the algorithm. To examine the resolution of the proposed method, we compared the localization result using the pairwise off-grid BSBL with other beamformers. Synthetic data, measured from two adjacent monopole noise sources, was exploited to analyze the resolution of the pairwise off-grid BSBL in Section 4.1. Subsequently, cavitation experimental data involving two adjacent impulsive cavitations is applied to the proposed method to analyze the localization performance in the actual environment (Section 4.2 and Section 4.3). Compared with other beamformers, we will show that the 3D pairwise off-grid BSBL can provide high-resolution localization performance with a moderate computational burden.

### 4.1. Localization of Synthetic Noise Sources

Numerical tests were examined to validate the localization performance. Figure 5 shows the numerical environment for the localization process which involves the two adjacent monopole sources emitting the spherical waves. We set the search space (𝑢-axis: −0.05 to 0.15 m; 𝑣-axis: −0.10 to 0.10 m; 𝑤-axis: −0.03 to 0.17 m) and allocate the simulated sources, having the same amplitude at vertical plane 𝑢 = 0.00 m: (0, 0.03, 0.01) m, (0, 0.06, 0.04) m. For the localization, 13 frequency components ([20:5:80] kHz) are measured using the 6 hydrophones (*L* = 13 and *M* = 6) and 1 s long time signals are recorded with a sampling frequency of 256 kHz. The signal-to-noise ratio (SNR) in the tests was set to 20 dB, which was sufficient to discriminate the sources from noise. We used grid intervals varying with *d* = 0.025, 0.05, and 0.10 m for each localization process, then examined the resolution and computational burden of the localization process along with various beamformers.

Figure 6a shows the localization result using the conventional beamformer. Correlations between the measurements and replicas were displayed with ambiguous surfaces, and corresponding values were normalized with respect to the maximum value. Two ellipses containing the actual source positions are observed in these results; however, it incurs ambiguities due to the poor resolution, especially for the direction of the *w*-axis. Figure 6b shows the localization results using the on-grid BSBL [11], which ignores the modeling error owing to basis mismatch. Among the grid points (marked with a lattice), the five largest components from the on-grid BSBL are projected onto the *v*–*w* plane, and two major components were observed at the true source positions. In this result, we adopted the grid distribution matching for the true source locations; thus, there was no modeling error owing to basis mismatch. Without the modeling error, on-grid BSBL estimated the exact source locations in this case.

Meanwhile, basis mismatch significantly degrades the localization performance when using the on-grid BSBL. Figure 7 shows the grid distributions (marked with a lattice) and the source positions (marked with ‘x’) where basis mismatch is present (grid intervals are set to 0.10, 0.05, and 0.025 m, respectively). Under the basis mismatch, on-grid BSBL estimates the locations irrelevant to actual source positions regardless of grid intervals. Even though the dense grid is exploited (Figure 7c), the on-grid approach suffers from the error due to basis mismatch.

Off-grid approaches are adopted to reduce performance degradation due to basis mismatch. Figure 8a–c show the localization results using the off-grid BSBL with grid intervals 0.10, 0.05, and 0.025 m, respectively. By adopting the off-grid approach, modeling error induced by basis mismatch was significantly reduced, and one dominant component was commonly observed near the actual source position. However, only the result using the finest grid interval (0.025 m) exhibited the source separation performance and suffered significant computational complexity (approximately 50 min). Otherwise, pairwise off-grid BSBL (Figure 8d) accurately separated the adjacent sources with a moderate grid interval (0.10 m) and significantly decreased the computational burden (approximately 29 s). For clarity, the simulation results are summarized in Table 1.

### 4.2. Localization of Transducer Source

Transducer data was applied to 3D pairwise off-grid BSBL in this section. Figure 9a illustrates the configuration of the transducer experiment conducted in the cavitation tunnel. One transducer source (ITC-1001) with a known position ([u,v,w]=[0.0, 0.348, 0.0] m) was emitting the broadband spherical waves over the frequencies from 7 kHz to 37 kHz, and corresponding acoustic data was measured on six hydrophones (B&K 8103) mounted at the top of the cavitation tunnel. For the localization, 16 frequency components (L=16, [7:2:37] kHz) were obtained from the 1 s long acoustic signal using the fast Fourier transform (FFT), and 189 grid points ($N=64,N^{\prime }=125)$ at intervals of 0.10 m are exploited to set the search space (𝑢-axis: −0.20 to 0.20 m; 𝑣-axis: −0.60 to −0.20 m; 𝑤-axis: −0.20 to 0.20 m).

Figure 9b shows the localization results for the experiment, which are projected onto the 𝑢–𝑤 (looking from the portside) and 𝑣–𝑤 (looking from the stern side) planes for visualization. By using the pairwise off-grid BSBL, a single dominant component (colored square at [u,v,w]=[0.002 m,−0.335 m,−0.003 m]) is observed, which is very close to the actual position (marked with ‘x’).

### 4.3. Localization of Cavitation Noise Source

Spare reconstruction methods [7,8,9] outperformed traditional beamformers [3,4,5,6] (such as conventional and adaptive beamformers [34]) for cavitation localization with respect to resolution. However, these suffer from modeling error due to basis mismatch. Although the coherent frequency processing [10] was applied to the problems to mitigate the basis mismatch error, significant modeling errors still exist owing to basis mismatch. To overcome the basis mismatch, an off-grid spare reconstruction method (off-grid BSBL) was applied to the TVC localization problem [11] and exhibited high-resolution localization results despite the presence of basis mismatch. Although this scheme is effective in reducing the basis mismatch error, it requires a fine grid interval, which leads to a significant increase in computational complexity. In this section, we applied the pairwise off-grid BSBL to actual TVC data to estimate their accurate positions without a significant increase in computational burden.

Figure 10a shows a 0.1 s long signal (signal 3), including two impulse-type cavitations (signals 1 and 2) with a sampling frequency of 256 kHz, measured on the six hydrophones, as shown in Figure 10c. This short sustained and intermittent cavitation is categorized as pop-type cavitation and mostly occurs near the upper part of the propeller in the incipient stage (Figure 10b) [3,5]. From the signal, we extracted three snapshots (signals 1–3, as shown in Figure 10a) to generate the multiple measurement vector. Signals 1 and 2 have 2048 data points, including each pop-type cavitation. Meanwhile, signal 3 has 26,214 data points, including both pop-type cavitations (for the detailed experimental setup and the raw acoustic data, readers are referred to Appendix B). For the localization, zero padding was conducted to increase the resolution, and 13 frequency components ([20:5:80] kHz) were sampled to generate multiple measurements. The search space (𝑢-axis: −0.05 to 0.15 m; 𝑣-axis: −0.10 to 0.10 m; 𝑤-axis: −0.03 to 0.17 m) comprises the grid points with the grid interval 0.05 m for the localizations using BSBLs.

Figure 11 and Figure 12 display the localization results of signals 1 and 2 for the conventional beamformer, the off-grid BSBL, and the pairwise off-grid BSBL. The conventional beamformer estimated the TVC location with the ambiguity surface on the *v–w* and *u–w* planes, which had a maximum value of 1 (Figure 11a and Figure 12a). One dominant ellipse appeared near the top of the propeller, indicating the presence of one pop-type cavitation; however, the conventional beamformer still suffered from poor resolution along the vertical axis. On the other hand, the off-grid-based methods display the five largest components, which were projected onto the *u*–*w* and *u*–*w* planes and provide one dominant component representing the pop-type cavitation noise (Figure 11b,c and Figure 12b,c). For the measurement, including single pop-type cavitation, off-grid BSBLs eliminate ambiguity in the conventional beamformer and provide a distinct source position.

Figure 13 shows the pop-type cavitation localization results using the 0.1 s long time signal (signal 3) for the conventional beamformer and various BSBLs. Since two cavitation signals are highly correlated (or highly coherent), the conventional beamformer only shows the one dominant ellipse corresponding to the first cavitation event (Figure 13a). Figure 13b illustrates the localization results for the on-grid BSBL, which result in the components at the positions irrelevant to the cavitation events. Although a fine grid interval (=0.025 m) is adopted in this process, on-grid BSBL fails to estimate cavitation positions due to basis mismatch. Meanwhile, off-grid BSBL, using a moderate grid interval (=0.05 m), was applied to the signal and displayed the one dominant component, corresponding to the first cavitation event (Figure 13c). Although off-grid BSBL estimated the acoustic centers for the cavitations, it failed to distinguish two noise sources due to their adjacency. In contrast to other methods, pairwise off-grid BSBL presents two dominant components which represent two cavitation events and achieves outperformed localization results (Figure 13d). Even though two pop-type cavitations are adjacent, the proposed method provides proper representation and the most precise localization result among the methods.

By adopting the pairwise grid sets, off-grid localization problems, such as off-grid 1D and 2D DOA estimations [25,26,27,28,29], can be extended to a pairwise off-grid model, which may enhance the computational efficiency as well as resolution of the localization result. In a future study, we plan to extend the pairwise off-grid BSBL to 1D DOA estimations. Moreover, we will examine the performance of the proposed method along with the various cavitation noise sources (such as sheet, root, and pocket cavitations).

## 5. Conclusions

We proposed a computationally efficient 3D off-grid BSBL method that can improve the localization performance for the adjacent cavitation noise sources. Unlike the conventional off-grid BSBL method, which utilizes a uniform grid set, our method considered two different grid sets with a moderate grid interval. To the best of our knowledge, this study is the first attempt to express the off-grid cavitations using the different grid sets. The pairwise off-grid scheme exhibited several advantages. Firstly, it efficiently reduced the modeling error regardless of grid interval, which, in turn, enhanced the localization performance. Secondly, it significantly reduced the computational complexity due to moderate grid interval usage.

The proposed method was examined by using the synthetic and experimental data along with other beamformers. Compared to the results from the other beamformers, 3D pairwise off-grid BSBL showed the high-resolution localization result for the off-grid cavitation even though the cavitation sources are adjacent, significantly reducing the computational complexity.

## Figures and Tables

**Figure 1 sensors-23-02628-f001:**
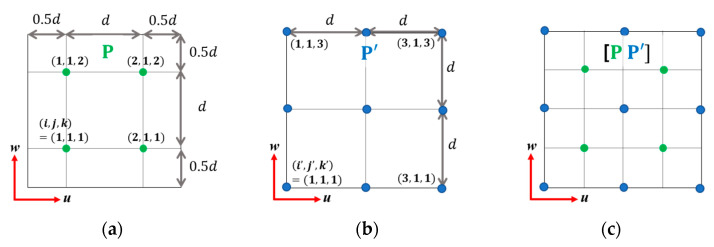
Two different grid sets and corresponding pairwise grid set projected on the *u–w* plane. (**a**) Distribution of the first grid set **P**; (*i, j, k*) represent the directional index of the first grid set for each *u*, *v*, and *w*-axis, respectively. (**b**) Distribution of the second grid set **P**’; (*i’, j’, k’*) represent the directional index of the second grid set for each *u*, *v*, and *w*-axis, respectively. (**c**) Pairwise grid set [**P P**’] in the search space comprising the first and second grid set.

**Figure 2 sensors-23-02628-f002:**
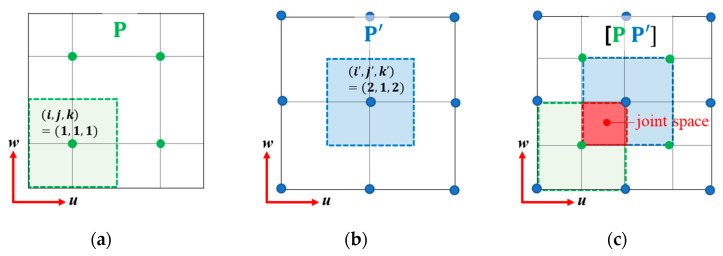
The *u–w* plane in the search space including two different grid sets. (**a**) (*i, j, k*)th grid point among the first grid set, and corresponding bound (marked with dashed line) and volume space (marked with colored square). (**b**) (*i’, j’, k’*)th grid point among the second grid set, and corresponding bound (marked with dashed line) and subspace (marked with colored square). (**c**) Joint space induced by the (*i, j, k*)th and the (*i’, j’, k’*)th grid points.

**Figure 3 sensors-23-02628-f003:**
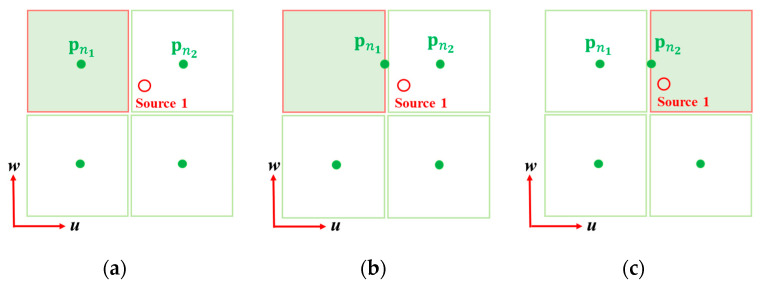
Grid activation and transfer process. (**a**) $p_{n_{1}}$ (Left-top grid) is initially located at $p_{n_{1}}^{\left(0\right)}$ and the grid is activated. (**b**) After some iteration, $p_{n_{1}}$ approaches to the off-grid cavitation and is captured at $u_{n}{}^{+}$. (**c**) $p_{n_{1}}$ is transferred to $p_{n_{2}}$(Right-top grid) and the $p_{n_{1}}$ is inactivated. $p_{n_{2}}$ inherits the $p_{n_{1}\;}$ and the grid is activated.

**Figure 4 sensors-23-02628-f004:**
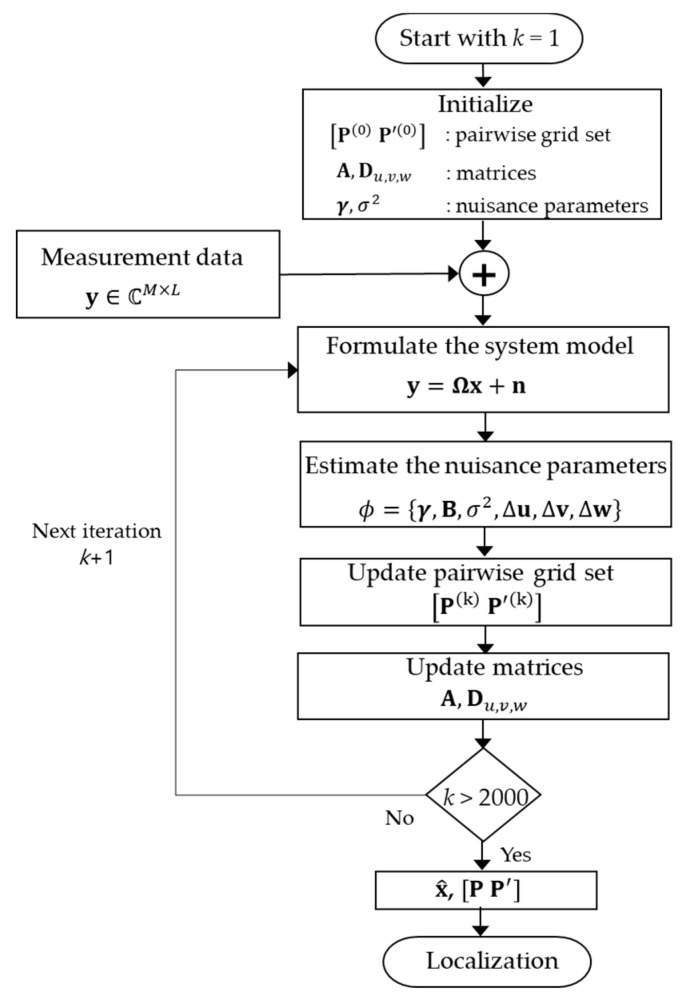
Flow-chart of 3D pairwise off-grid BSBL.

**Figure 5 sensors-23-02628-f005:**
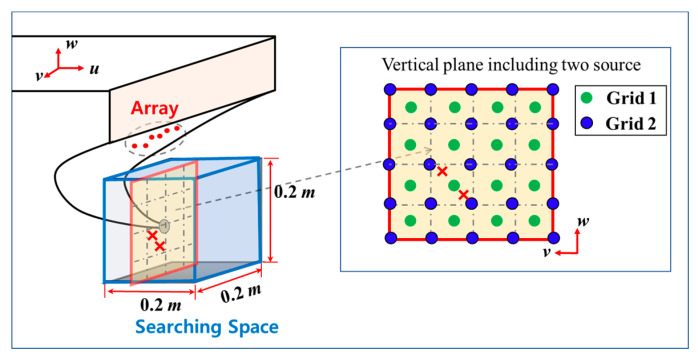
Numerical environment for examining the proposed localization method. Two adjacent noise sources were simulated emitting spherical sound waves, which were measured by an array located at the upper side of the search space. The amplitudes of the noise sources were set as the same on purpose in order to inspect the resolutions of considered schemes conveniently.

**Figure 6 sensors-23-02628-f006:**
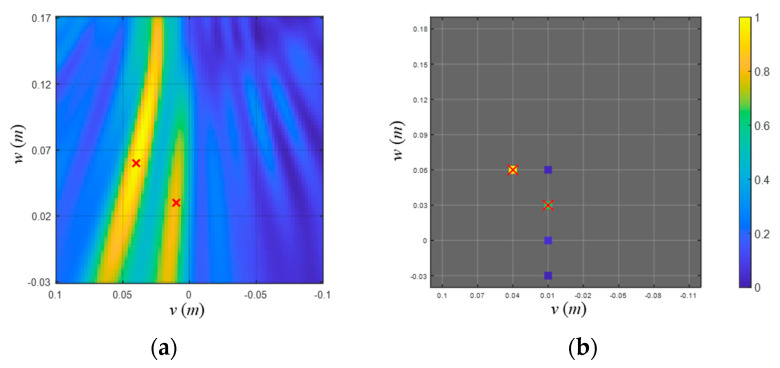
(**a**) Localization results using conventional (or Bartlett) beamformer. Ambiguity surfaces of the *v–w* (looking from the stern side) planes including two sources (marked with ‘x’) are displayed. The result was normalized with respect to the maximum value among the correlations. (**b**) Localization results using on-grid BSBL without basis mismatch. The five largest components (marked with colored squares) from the on-grid BSBL and two sources (marked with ‘x’) are projected onto the *v*–*w* (looking from the stern side) plane to visualize the 3D localization results. The lattice denotes the grid positions, and the results were normalized with respect to the maximum value among the components. The figure legend indicates the value of the components, which are normalized with respect to the maximum value.

**Figure 7 sensors-23-02628-f007:**
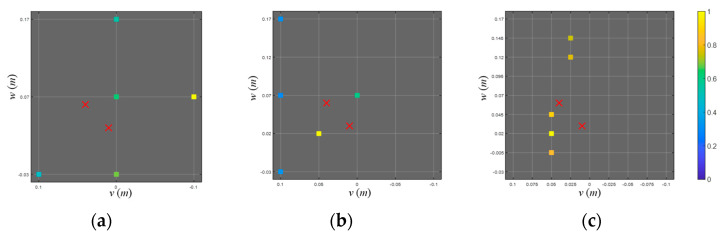
(**a**–**c**) Localization results using on-grid BSBL with grid intervals 0.10 m, 0.05 m, and 0.25 m. The five largest components (marked with colored squares) and two sources (marked with ‘x’) are projected onto the *v–w* (looking from the stern side) planes, and each case involves a modeling error owing to basis mismatch. Computational times measured from the localization process were approximately 7 s, 44 s, and 9 min, respectively, with a CPU i9-10900K.

**Figure 8 sensors-23-02628-f008:**
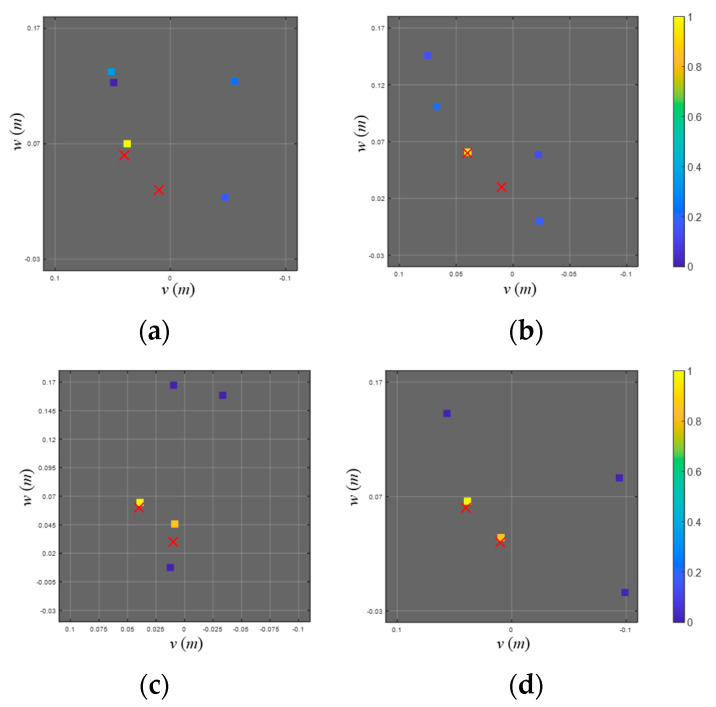
(**a**–**c**) Localization results using the off-grid BSBL with grid intervals 0.10 m, 0.05 m, and 0.25 m. Computational times measured from the localization processes were approximately 9 s, 56 s, and 50 min, respectively, with a CPU i9-10900K. (**d**) Localization results using the pairwise off-grid BSBL with a grid interval 0.10 m. Computational time measured from the localization process was approximately 29 s.

**Figure 9 sensors-23-02628-f009:**
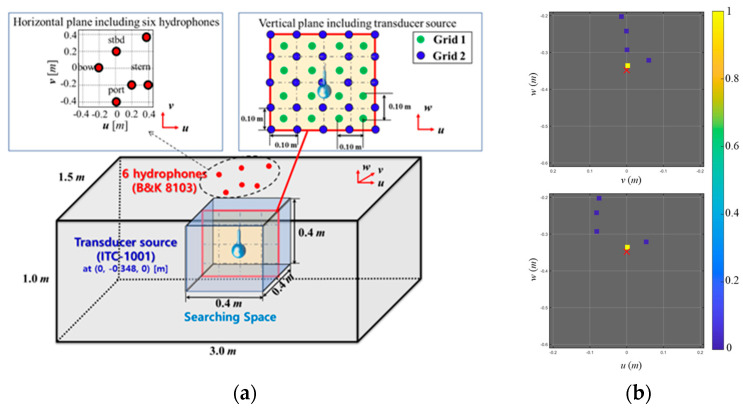
(**a**) Configuration of the transducer experiment, and (**b**) localization result using the pairwise off-grid BSBL. The five largest components were projected onto the 𝑢–𝑤 (looking from the port side) and 𝑣–𝑤 (looking from the stern side) planes. A single dominant component represented the transducer position at (0.002, −0.335, −0.003) m.

**Figure 10 sensors-23-02628-f010:**
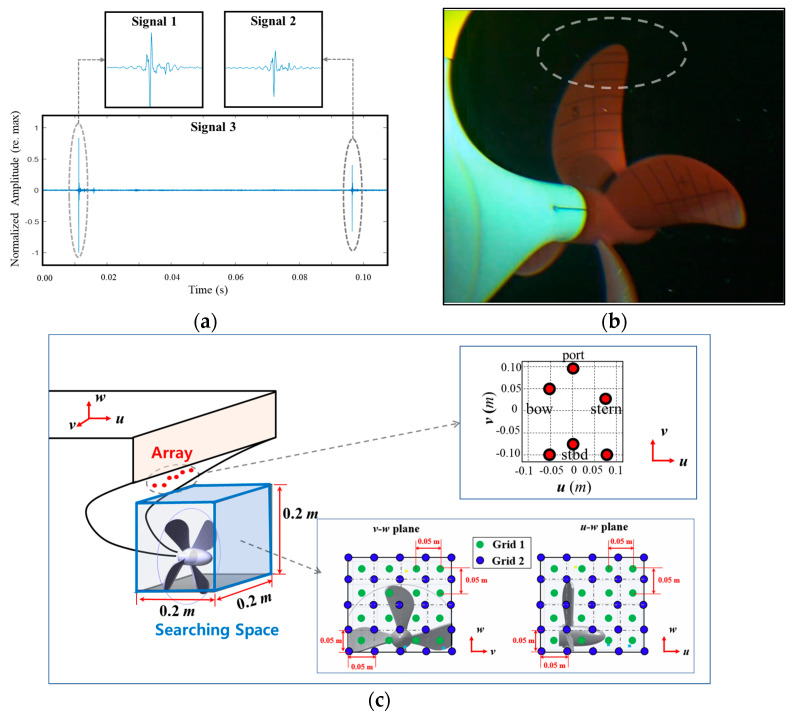
(**a**) Measured time signal involving two different impulsive cavitation noises. (**b**) Visual observation of the cavitation tunnel experiment. The rate of revolution of the propeller, and cavitation number were set as $0.65\times 10^{5}$ Pa, 1390 rev/min, and 4.12, respectively. (**c**) Configuration of six hydrophones and the search space for TVC localization.

**Figure 11 sensors-23-02628-f011:**
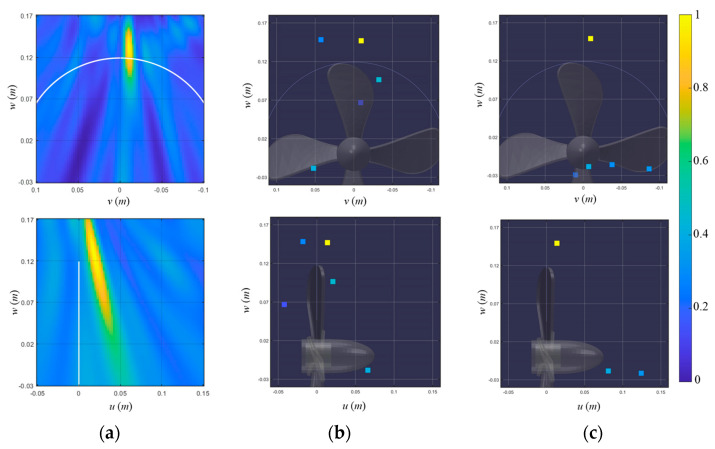
Localization results for the first cavitation event (signal 1) using (**a**) a conventional beamformer, (**b**) off-grid BSBL, and (**c**) pairwise off-grid BSBL. The localization results are displayed onto the 𝑣–𝑤 (looking from the stern side) and 𝑢–𝑤 (looking from the port side) planes to visualize the 3D cavitation localization results.

**Figure 12 sensors-23-02628-f012:**
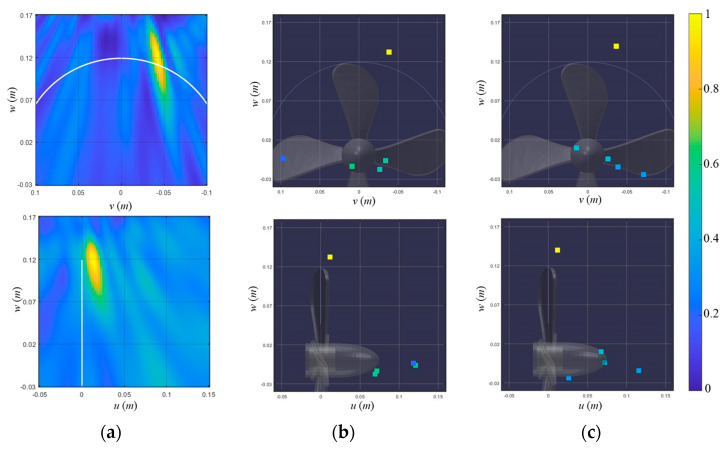
Localization results for the second cavitation event (signal 2) using (**a**) conventional beamformer, (**b**) off-grid BSBL, and (**c**) pairwise off-grid BSBL. The localization results are displayed onto the 𝑣–𝑤 (looking from the stern side) and 𝑢–𝑤 (looking from the port side) planes to visualize the 3D cavitation localization results.

**Figure 13 sensors-23-02628-f013:**
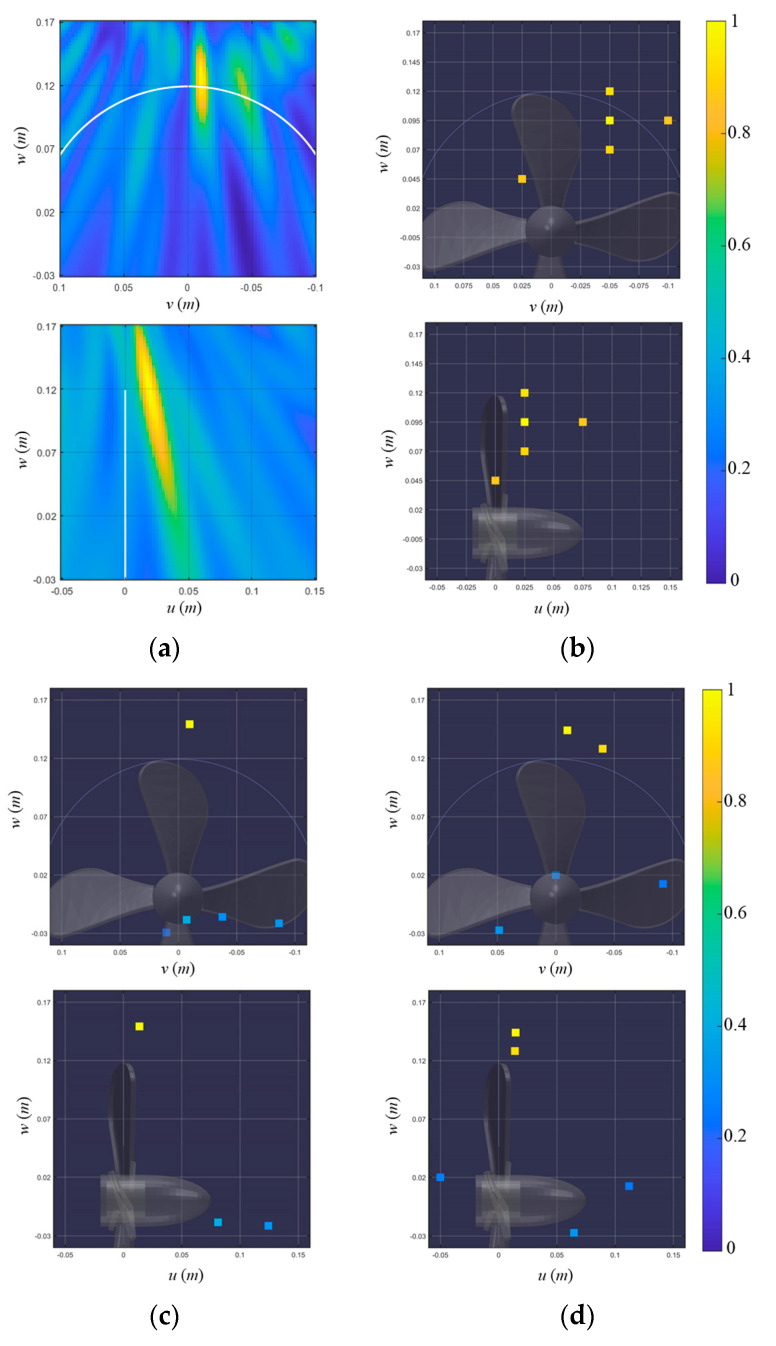
Localization results of signal 3, which include the first and second cavitation events, using (**a**) a conventional beamformer, (**b**) on-grid BSBL, (**c**) off-grid BSBL, and (**d**) pairwise off-grid BSBL. The localization results are displayed onto the 𝑣–𝑤 (looking from the stern side) and 𝑢–𝑤 (looking from the port side) planes to visualize the 3D cavitation localization results.

**Table 1 sensors-23-02628-t001:** Localization results of adjacent noise sources along with various schemes.

	Grid Interval (m)	Separation Ability	Computational Time (s)	Localization Performance
CBF	-	O	21	Two source positions are estimated ambiguously. (Low resolution along with the vertical axis)
Off-grid BSBL	0.100	X	9	High resolution. Two sources could not be separated.
	0.050	X	56	
	0.025	O	2923	High resolution. Two sources are distinguished.
Pairwise off-grid BSBL	0.100	O	29	

The grid interval does not affect the localization performance of CBF. Computational times were measured with a CPU i9-10900K.

## Data Availability

Not applicable.

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
