# Peer review of "3D Off-Grid Localization for Adjacent Cavitation Noise Sources Using Bayesian Inference"

_sensors, 2023, doi:10.3390/s23052628_

Round 1
Reviewer 1 Report
Plz see the attachment.

Reviewer 2 Report
This work describes a sparse localization method for off-grid cavitation to estimate their precise locations while keeping reasonable computational efficiency. It adopts two different grid sets with a moderate grid interval and provides redundant representations for adjacent noise sources. The logic is clear and the findings are meaningful. This paper can be published after minor revision, the following are some suggestions for the revision:
1. The abstract needs more quantitative results. The abstract section is an important and powerful representation of the research. It is better that the results should be presented with the support of specified data. Please provide your contribution and work novelty.
2. Relevant references should be provided for all equations.
3. The authors should indicate this work to enhance system performance. Also, the author should add more references that discuss the effect of using this work. It is recommended that the authors carry out wide analysis and comparison with the state-of-the-art studies.
4. The conclusion section on lacks in summative conclusions. The main results, novelty and academic contributions should be emphasized in this section. Moreover, are the results obtained in this paper really applicable in other similar researches?
Reviewer 3 Report
The author proposed a sparse localization method for off-grid cavitation, which was applied to simulated and experimental data to examine the performance compared with the conventional localization method. The content of this paper has certain innovation and application value, however, there are also some noteworthy problems.
1. The first sec5 of line 77 should be changed to sec4.
2. In line 80, the author describes that TVC will cause broadband noise, which is true in practice. However, in the simulation/experiment examples in the following Results, the author only performs cavitation localization at multiple single frequencies. Can the method proposed in this paper locate the results of continuous broadband cavitation noise?
3. In the Results, can the author show some original data before processing with this method? At present, only the results of cavitation localization are shown, which is not very intuitive.
4. Will the content in 3.5 be more clearly expressed in the form of flow chart?
5. The numerical example described in lines 330-332 contains 13 frequency components. The author should explain them in more detail, for example, what are the frequencies? Are the frequencies swept one by one or the signals of different frequencies synthesized in the calculation?
6. In the results and analysis, similar to Figure 6, the comparison of the same simulation/experimental results at different grid resolutions or using different methods for cavitation localization are only shown by figures. Adding some quantitative analysis will make the conclusion more intuitive.
7. Figure 9 shows an experiment setup of flow noise around a propeller. In this experiment, what is the propeller's operating state (advancing state)? The model and basic parameters of the propeller used in the experiment, the size of the basin also need to be supplemented.
Reviewer 4 Report
Title: 3-D Off-grid Localization for Adjacent Cavitation Noise Sources Using Bayesian Inference
This manuscript introduced a localization method for off-grid cavitation that utilizes pairwise grid sets to estimate the cavitation positions with improved computational efficiency. The writing is generally clear. However, there are a few specific comments that need to be addressed:
1. In the experiment, the pairwise off grid BSBL method are tested in a specific environment. The authors should address the generalizability of their method to different environments and noise sources, as well as its applicable range.
2. In section 4, the meaning of the color bar should be explained.
3. Figure 7d, x and y axis titles need to be justified.
4. The authors should provide a more quantitative evaluation of the localization results, such as a comparison with actual TVC data, to demonstrate the advantages of their pairwise off-grid BSBL method for detecting cavitation position.
Round 2
Reviewer 1 Report
Now it can be accepted for publication.